# Fabrication of PbO_2_ Electrodes with Different Doses of Er Doping for Sulfonamides Degradation

**DOI:** 10.3390/ijerph192013503

**Published:** 2022-10-19

**Authors:** Tianyu Zheng, Chunli Wei, Hanzhi Chen, Jin Xu, Yanhong Wu, Xuan Xing

**Affiliations:** Department of Environmental Science, College of Life and Environmental Science, Minzu University of China, Beijing 100081, China

**Keywords:** PbO_2_ electrode, Er doping, electrochemical oxidation, sulfamerazine, intermediates

## Abstract

In the present study, PbO_2_ electrodes, doped with different doses of Er (0%, 0.5%, 1%, 2%, and 4%), were fabricated and characterized. Surface morphology characterization by SEM-EDS and XRD showed that Er was successfully doped into the PbO_2_ catalyst layer and the particle size of Er-PbO_2_ was reduced significantly. Electrochemical oxidation of sulfamerazine (SMR) in the Er-PbO_2_ anode system obeyed te pseudo first-order kinetic model with the order of 2% Er-PbO_2_ > 4% Er-PbO_2_ > 1% Er-PbO_2_ > 0.5% Er-PbO_2_ > 0% PbO_2_. For 2% Er-PbO_2_, *k_SMR_* was 1.39 h^−1^, which was only 0.93 h^−1^ for 0% PbO_2_. Effects of different operational parameters on SMR degradation in 2% Er-PbO_2_ anode system were investigated, including the initial pH of the electrolyte and current density. Under the situation of an initial pH of 3, a current density of 30 mA·cm^−2^, a concentration of SMR 30 mg L^−1^, and 0.2 M Na_2_SO_4_ used as supporting electrolyte, SMR was totally removed in 3 h, and COD mineralization efficiency was achieved 71.3% after 6 h electrolysis. Furthermore, the degradation pathway of SMR was proposed as combining the active sites identification by density functional calculation (DFT) and intermediates detection by LC-MS. Results showed that Er-PbO_2_ has great potential for antibiotic wastewater treatment in practical applications.

## 1. Introduction

Antibiotics have been commonly used in the therapy of human diseases and the development of animal husbandry [1]. Global antibiotic use, which was only 22.1 DDDs (defined daily doses) in 2000, had reached 34.8 billion DDDs by 2015 [2]. What is more, antibiotics cannot be completely metabolized and some of them are excreted by feces and urine. Based on Ying Guangguo’s study, about 53,800 tons of antibiotics in China are discharged into the surroundings each year and the average concentration value is up to 303 ng L^−1^ in surface water [3]. The massive proliferation of antibiotics threatens environmental safety and human health even at an occasional level, by inducing resistance mechanisms on microorganisms. In addition, the persistent presence of organic pollution will catalyze the generation of superbugs, posing a great threat to human health [4]. Among all the antibiotics, sulfamerazine (SMR) is one amongst the foremost well-liked used antibiotics in animal treatment [5]. Therefore, an effective and efficient degradation method for SMR removal is urgently required.

Many technologies have been applied for antibiotics degradation, such as adsorption [6], ozone oxidation [7], photo-catalyst oxidation [8], persulfate oxidation [9] and electrochemical oxidation [10]. Among all ways, electrochemical oxidation has attracted increasing consideration because of its simple manipulation, the mild conditions of the process, lack of secondary pollution, and high oxidation power [11,12,13]. The anode is the key issue for electrochemical oxidation process. Among all the electrode materials, PbO_2_ electrode has been widely used because of its low price, high oxygen evolution potential, strong oxidation, and corrosion-resistance ability [14,15,16].

The doping method has been applied to improve the oxidation ability of PbO_2_ electrodes [17,18] and many kinds of materials have been investigated, including metal ions (Bi^3+^, Fe^3+^, Co^2+^) [19], non-metallic ions (F^−^) [20], metal oxide particles [21], and surface active substances of polytetrafluoroethylene (PTFE) [22]. Among them, rare earth elements are widely used because of their special 4f orbital structure and excellent catalytic properties in catalytic modification [23]. Four kinds of rare earth elements (Gd, La, Nd, Ce)-doped PbO_2_ electrodes were prepared by Shen Hong [24,25] and the results demonstrated that different rare earth elements have different catalytic effects on the electrodes. As a typical rare earth lanthanide, Er has a large ionic radius and a stable electron saturation structure, which makes it an advantageous doping material. For example, Wang Ying [26] designed Er-chitosan-PbO_2_ electrode for degradation of 2,4-DCP. Li Shuanghui [27] introduced Er^3+^ ions into SnO_2_ lattice to synthesize Er^3+^-SnO_2_ nanobelts via thermal evaporation methodology, which obviously improved the sensing performance of SnO_2_ electrodes. Wang Yanping [28] prepared Ti/SnO_2_-Sb/Er-PbO_2_ electrodes for the electrochemical degradation of SMX. However, the electrodes were not characterized and the effect of Er doping amount was not discussed. Zhou Yuanzhen [29] investigated the electrochemical degradation performance for methylene blue based on the novel step-to-step fabricated Ti/Sb_2_O_3_-SnO_2_/Er-PbO_2_ anodes. On the basis of this study, we added a discussion on the amount of Er incorporation and changed the pollutant from dye to SMR, which is more difficult to degrade. In summary, Er doped in PbO_2_ electrode has not been investigated comprehensively, and the research limitation should be remedied.

In this work, Er-PbO_2_ electrode was fabricated (Ti/SnO_2_/*α*-PbO_2_/Er-*β*-PbO_2_) with SnO_2_ [30] and an *α*-PbO_2_ [28] layer was inserted between the Ti substrate and *β*-PbO_2_ layer to reduce interface resistance and extend service life. The Er-PbO_2_ electrode was characterized by X-ray diffraction (XRD) and scanning electronic microscopy with energy dispersive spectroscopy (SEM-EDS) systematically. At the same time, effects of current density and pH value on the degradation of SMR using the prepared Er-PbO_2_ electrode were researched. In addition, the possible degradation pathways of SMR were analyzed by density function theory (DFT) calculation and LC-MS intermediates detection. This work aims to develop an efficient and green SMR treatment system that can be used to prevent SMR from spreading pollution and threatening human health.

## 2. Experimental

### 2.1. Chemicals

All chemical reagents employed in the experiments were of analytical quality. Acetone, lead oxide, sodium hydroxide, lead nitrate, potassium dichromate, silver sulfate, and mercury sulfate were purchased from traditional Chinese Medicine Group Chemical Reagent Co., Ltd., Beijing, China. Ammonium fluoride, hydrofluoric acid, nitric acid, phosphate, concentrated acid, and n-butanol were purchased from Beijing Chemical Plant, Beijing, China. Antimony trichloride were bought from Tianjin Dachen Chemical Reagent, Tianjin, China. Tin tetrachloride pentahydrate and sodium fluoride were bought from Tianjin Fuchen Chemical Reagent Plant, Tianjin, China. Ethylene glycol was obtained from Tianjin Zhiyuan Chemical Reagent Co., Ltd., Tianjin, China. Nitrate bait was obtained from Shaanxi Ruikexin material Co., Ltd., Shaanxi, China. Polytetrafluoroethylene was obtained from Dongguan Jianyang Polymer material Co., Ltd., Guangdong, China. Sulfamerazine was obtained from Beijing Solaibao Technology Co., Ltd., Beijing, China. Ultrapure water from a Millipore Milli-Q system (>18 mΩ cm^−1^) was used for all the solutions preparation at 25 ± 1 °C.

### 2.2. Electrode Preparation

Titanium sheets (20 mm × 15 mm× 1.5 mm) were polished with 240 mesh, 320 mesh, and 600 mesh sandpaper successively and ultrasonically cleaned in deionized water and acetone for 15 min, then the titanium sheets were etched in the solution with a HF:H_2_O:HNO_3_ ratio of 1:5:4. After pretreatment, using the etched Ti plate as an anode and the noble metal sheet as cathode, with electrode spacing of 1.5 cm, at 30 V constant pressure, the electrode was electrolyzed in 2 V% H_2_O and 0.3 wt% NH_4_F ethylene glycol solution for 0.5 h and in ethylene glycol solution containing 5 wt% H_3_PO_4_ for 1 h. Then the titanium nanotube was prepared after sintering at a rate of 5 °C min^−1^ to 450 °C in a tube furnace.

The titanium nanotube was brushed with the solution containing 3.2 g SbCl_3_ and 15 g SnCl_4_-5H_2_O in 5 mL concentrated hydrochloric acid and 30 mL n-butanol at 25 °C, and then dried in an oven at 145 °C for 20 min. This step was repeated 6 times and then the electrode was baked in a tubular furnace at 500 °C for 2 h. All the operations were repeated once, and the SnO_2_-Sb_2_O_3_ mixed oxides on the face of titanium plate were procured after cooling.

The Ti/SnO_2_-Sb substrate was electrodeposited with the middle layer of *α*-PbO_2_ in basic solution (0.1M PbO, 4M NaOH) at 50 °C and application of ampere density of 5 mA cm^−2^ for 1 h. Lastly, pure and doped *β*-PbO_2_ membranes were electrodeposited on the *α*-PbO_2_ middle layer in acidic solution at 65 °C, applying ampere density of 50 mA cm^−2^ for 1.5 h. The synthetic acidic solution consisted of 0.5 M Pb(NO_3_)_2_, 0.05 M NaF, and 1.8 mL PTFE in 1 M HNO_3_. When the Ti/Sb_2_O_3_-SnO_2_/Er-PbO_2_ anodes were prepared, 2.5 mM, 5 mM, 10 mM, and 20 mM of Er(NO_3_)_3_ •5H_2_O were supplemental into the acidic electroplating answer for the deposit, separately.

### 2.3. Electrode Characterization

In this experiment, the element composition and surface morphology of the electrodes were inspected by SEM-EDS (Japanese Hitachi s4800, Hitachi Limited, Tokyo, Japan). The crystal phase composition and the grain size were analyzed by XRD, which was XD-DI type, using Cu Kα radiation (36 KV, 30 mA). Scanning speed was 4°min^−1^, 2θ = 20°–80°.

### 2.4. Electrocatalytic Degradation of SMR

The electrocatalytic degradation experiments of SMR were carried out under galvanostatic conditions in a 500 mL beaker with a magnetic stirrer. The prepared Er-PbO_2_ electrode served as the anode and a stainless-steel electrode of the uniform dimension served as the cathode. They were placed vertically and parallel to one another at an interval of 1.5 cm. The initial concentration of SMR was 30 mg L^−1^, and 0.2 M Na_2_SO_4_ was used as a supporting electrolyte. During the experiments, samples (10 mL) were taken from the reactor at specific time intervals and then analyzed for SMR concentration and chemical oxygen demand (COD). The reaction temperature was maintained at 25 °C throughout all the experimental runs.

### 2.5. Analytical and Calculation Methods

The linear sweep voltammetry experiment was executed to acquire their oxygen evolution overpotential at room temperature using a computerized electrochemical workstation (CHI 630E, Shanghai Chenhua, Shanghai, China) with a conventional three-electrode system, where the prepared electrodes served as working electrodes, while a platinum sheet and a saturated calomel electrode (SCE) were used as the counter and reference electrodes, respectively. The concentrations of SMR were measured by high performance liquid chromatography system (HPLC, LC-20A, Shimadzu Company, Kyoto, Japan). The mobile liquid phase was a mixed solution with 60% (by volume) methanol and 40% water. The separation was implemented using an Agilent SB-C18 (4.6 mm × 250 mm, 5 μm) at a pillar temperature of 25 °C and at a flow rate of 1 mL min^−1^, The UV detector wavelength was set at 265 nm. The injection volume was 25 μL.

COD was determined according to the national standard method (GB11914-1989). The current efficiency was computed as Equation (1) [31]:(1)$$CE=\frac{COD_{0}-COD_{t}}{8It}FV\times 100\%$$
where *COD**_t_* and *COD_0_* are the chemical oxygen demand (g L^−1^) at time *t* (s) and zero, separately, *I* is the current (A), *t* is the electrolysis time (h), *F* is the Faraday constant (96,287 C mol^−1^), and *V* is the electrolyte volume (L).

The energy consumption (*E_C_*) in the process of electrochemical oxidation was computed using Equation (2) [32]:(2)$$E_{C}=\frac{UIt}{1000V}$$
where *U* is the average cell voltage (V), *I* is the current (A), *t* is the degradation time (h), *V* is the wastewater volume (m^3^).

The intermediates during the degradation of SMR were determined with LC-MS (LC-20ADXR, Shimadzu, Tokyo, Japan and API3200 Qtrap, Applied Biosystems, Waltham, Mass, USA). The mobile phase consisted of two solutions, namely, A and B. Solution A was high pure water containing 0.01% formic acid, whereas solution B was methanol. The flow velocity was 0.4 mL min^−1^ and the temperature was kept at 40 °C.

The active point of SMR in the degradation process of electrocatalytic system was inferred by DFT calculation, and the related work was carried out by the Gaussian 09 program. Optimization of SMR geometry at the atomic level of the DFT theory B3LYP/6-31G (d, p). Afterwards, the active sites of SMR molecules vulnerable to free radical attack were identified by calculating the Fukui function (Equation (3)).
(3)$$f_{k}^{0}=[(q_{k}N-1)-(q_{k}N+1)]/2\;$$

## 3. Results and Discussion

### 3.1. Characteristics of Er-PbO_2_ Anodes

#### 3.1.1. Life Comparison between Titanium Nanotubes and Titanium Plates

According to previous research [33], the Ti plate is oxidized into nanotubes (nanocrystalline coral-like TiO_2_, TiO_2_-NCs) substrate before deposition of catalytic layer (Appendix A). The accelerated service life of TiO_2_-NCs/Sb-SnO_2_/PbO_2_ has been compared with Ti/Sb-SnO_2_/PbO_2_, in which Ti plates are used without any treatment. Results in Appendix A showed that the accelerated life time is 10 h and 25 h and the service life time is 2.85 y and 9.99 y, respectively. These results indicate that the conversion of Ti-based oxygen to TiO_2_-NCs can effectively extend the lifetime of the electrode.

#### 3.1.2. SEM-EDS Analysis of Er-PbO_2_ Anodes

The elemental morphology and composition of the prepared Er-PbO_2_ anodes were analyzed by SEM-EDS (Figure 1 and Figure 2, respectively). Results showed that a large number of cracks appeared on the electrode surface without Er doping (Figure 1a). These cracks would permit reactive oxygen species to penetrate into the titanium matrix during electrochemical oxidation process, passivate the Ti substrate, increase internal stress of inside electrode, and then cause the PbO_2_ layer to peel off. However, with a certain dose of Er doped into PbO_2_, the electrode surface is smooth and dense and the cracks are obviously reduced (Figure 1b–d) [34]. However, when the dose of Er is increased to 4%, the surface gap is enlarged again (Figure 1e), which would reduce the electro-catalytic activity [35]. In addition, EDS analysis of different electrodes showed that an Er element existed on the surface of electrodes (Figure 2) and weight and atomic percentage is shown in Appendix A, EDS results of different Er-PbO2 electrodes in Appendix A.

#### 3.1.3. XRD Analysis of Er-PbO_2_ Anodes

Diffraction peaks of XRD observed at 2θ of 25.3°, 31.8°, 36.1°, 49.0°, and 62.4° were assigned to (110), (101), (200), (211), (220), and (301) planes of β-PbO_2_, respectively (Figure 3). These results have a fine consistency with the standard data of the JCPDS card (number: 760564) [36,37]. However, diffraction peaks corresponding to rare earth elements Er did not appear after doping. In the present study, Er doped into PbO_2_ would replace the position of Pb^4+^ in the lattice. If the dose was not large enough, the lattice structure would change significantly, so that the doping could not be detected by XRD [38].

Crystal sizes of PbO_2_ electrodes with different dose of Er doping have been calculated according to the Scherrer equation (Table 1). This phenomenon could be attributed to the doping of Er introducing a fresh nucleation site for the growth of PbO_2_ crystals, which increased the quantity of crystal nuclei and hindered the expansion of particle size at the same time as the crystallization process [39]. Among them, the 2% Er-PbO_2_ had the smallest particle size (39.96 nm), while the particle size of undoped PbO_2_ was 43.43 nm. Smaller granularity size will give a bigger specific surface area, increase the active sites, and accelerate the reaction rate.

#### 3.1.4. OEP Analysis of Er-PbO_2_ Anodes

The curves of linear sweep voltammetry of Er-PbO_2_ with the sampling rate of 10 mV s^−1^ were investigated to examine the oxygen evolution potential (OEP) (Figure 4). The OEP increased in the order of 0% PbO_2_, 1% Er-PbO_2_, 4% Er-PbO_2_, 0.5% Er-PbO_2_, and 2% Er-PbO_2_ electrodes. It is obvious that Er doping enhanced the OEP of PbO_2_ electrode, which can shorten the aspect reaction of oxygen evolution, improve the current efficiency, and enhance the electrochemical oxidation ability [40].

### 3.2. Electrochemical Oxidation of SMR

#### 3.2.1. SMR Degradation of Er-PbO_2_ Anodes

The electrochemical oxidation of SMR utilization Er doped PbO_2_ electrodes was investigated with operating conditions of current density 30 mA cm^−2^, initial SMR thickness 30 mg L^−1^, and electrolyte Na_2_SO_4_ 0.2 M (Figure 5). The first-order kinetic fitting of the electrochemical decomposition process is shown in Equation (4).
ln(C_0_/C) = *k*t (4)
where C_0_ and C is the concentration of SMR (mg L^−1^) at electrolysis time of 0 and t (h), *k* was the reaction rate constant (h^−1^).

After 6 h electrolysis, the removal efficiency of SMR was 99% in all PbO_2_ electrode systems (Figure 5a). The plot of time (t) versus ln (C/C_0_) showed a straight line, reflecting that the oxidation reaction followed pseudo-first-order kinetics (Appendix A). The highest reaction rate constant of *k* with the value of 1.39 h^−1^ was obtained in 2% Er-PbO_2_ system. This value was higher than that in 4% Er-PbO_2_ (1.25 h^−1^), 1% Er-PbO_2_ (1.22 h^−1^), 0.5% Er-PbO_2_ (1.19 h^−1^) and 0% PbO_2_ (0.93 h^−1^) (Appendix A). Er-doped PbO_2_ electrodes showed excellent performance for SMR degradation, especially for 2% Er-PbO_2_, which is consistent with SEM and XRD detection results [41].

COD removal rates of 0% Er-PbO_2_, 0.5% Er-PbO_2_, 1% Er-PbO_2_, 2% Er-PbO_2_, and 4% Er-PbO_2_ were 57%, 71%, 73%, 86%, and 65% after 6 h electrolysis (Figure 5b). The mineralization ability order was the same with SMR degradation and 2% Er-PbO_2_ also showed excellent mineralization ability compared to other electrodes. The relationship between COD and SMR removal rate represented the accumulation of intermediate products during the degradation process [34]. The ratio of COD/SMR for 2% Er-PbO_2_ was 87%, indicating that the mineralization ability was strong with little accumulation of intermediates (Figure 5c).

#### 3.2.2. Effect of Current Density

The effects of current density on SMR degradation in 2% Er-PbO_2_ electrochemical oxidation system were investigated (Figure 6) [42,43]. Degradation efficiency of SMR and COD increased with current density increases. After 6 h degradation, the removal rates of SMR achieved 96.61%, 97.88%, and 98.11% and the first order pseudo-kinetic constant of *k* was 0.96, 1.33, and 1.42 with current densities of 10, 20 and 30 mA cm^−2^, respectively (Appendix A). The difference between COD removal was more significant compared with SMR. Degradation efficiency was 50%, 60%, and 71.4% with current densities of 10, 20, and 30 mA cm^−2^ after 6 h electrolysis, respectively.

Current efficiency (CE) (Figure 6c) and the energy consumption (E_c_) have also been analyzed (Figure 6d). Results show that along with electrolysis time extension, E_c_ increased while CE decreased [44]. When current density is 10 mA cm^−2^, CE was as high as 37.19% after 1 h reaction, but dropped to 14.87% after 6 h reaction. At the same time, CE decreased along with current density increasing. When the current density is increased from 10 to 30 mA cm^−2^, CE decreased from 14.87% to 5.78% after 6 h reaction. E_c_ increased along with current density rising and electrolysis time extension. When current density increased from 10 to 30 mA cm^−2^, E_c_ heightened from 1.52 to 6.25 kWh m^−3^. This result was mainly because of the increase of side reaction for oxygen evolution by •OH decomposition which induced energy waste during electrolysis process [13,45].

#### 3.2.3. Effect of pH Value

Effects of different pH value on SMR degradations were investigated (Figure 7). The removal rate of SMR and COD decreased when pH value increased from 3 to 11. The highest degradability was found at a pH value of 3, with the first order pseudo-kinetic constant *k* of 1.59 h^−1^, which was much higher than that of 1.38 h^−1^ and 0.63 h^−1^ under pH of 7 and 11 (Appendix A). The difference between COD removal was also significant under different pH conditions. After 6 h electrolysis, the efficiency of COD removal was as high as 95.4%, 84%, and 63.5% with pH of 3, 7, and 11, respectively.

This phenomenon was mostly because of the following two aspects. On one hand, •OH production was affected by pH condition, as shown in Equation (5). In acidic circumstances, H^+^ would restrain the dissolution of •OH into oxygen, which would improve the oxidation ability and current efficiency [46]
H_2_O → •OH + H^+^ + e^−^
(5)

On the other hand, morphology of SMR changes under different pH conditions because of the contents of functional groups such as aniline and heterocyclic alkali (Figure 8). The pKa_1_ value of SMR was 2.13 and the pKa_2_ value was 6.70 [47]. When the pH value of the solution is less than 2.13, SMR will be positively charged, which is not conducive to the detachment of N atoms from the aromatic ring. When the pH value of solution is higher than 6.70, the radical oxidation activity of aniline group will be enhanced [48,49].

### 3.3. Degradation Mechanism of SMR

#### 3.3.1. DFT Calculation

The Gaussian 09W calculation software was used in this study to optimize the geometry of SMR at the atomic level of DFT theory B3LYP/6-31G (d, p). The optimized structure and electron density distribution of SMR molecules are shown in Appendix A and Appendix A. The active site of SMR, expressed in terms of the Fukui index ($f_{k}^{0}$) for each atom, is shown in Table 2, with -OH attacking the atom with higher $f_{k}^{0}$ (Figure 9). Therefore, the •OH attack on C6, C12, O9, O10 and N7 may be the cause of SMR degradation.

#### 3.3.2. Intermediates Identification

Four intermediates of SMR shaped in electrochemical oxidation processes have been identified by LC-MS (Appendix A). The mass charge ratio (m/z), molecular formula, and molecular structure are summarized in Table 3. The results of LC-MS analysis showed that intermediates of T3 and T11 were formed by oxidation of C6 and N7 atoms. The atoms of C6 and N7 react preferentially with •OH due to their higher $f_{k}^{0}$ values. The production of T2 may be due to the high value of *f*_0_ for the reaction between •OH and O9 and O10 atoms. However, atoms such as C12 have high *f*
^0^ values, but it is difficult for them to be attacked by -OH due to the presence of spatial site resistance [48].

Based on the experimental results, a reasonable degradation principle of the electrochemical degradation of SMR has been proposed. As shown in Figure 10, the formation of T3 is formed by the hydroxylation of the sulfonamide bond. SMILEs rearrangement and fragmentation of S-N link result in the formation of T2. For the formation of T11, SMR was nitrified first (T9), then amidogen on the benzene ring was oxidized (T10). Finally, the methyl group on SMR was carboxylation (T11). Above all the aromatic intermediates were oxidized to carboxylic acid and eventually undergo mineralization to SO_4_^2^^−^, CO_2_, H_2_O, and NH_4_^+^.

## 4. Conclusions

In summary, PbO_2_ electrodes with different doses of Er doping were fabricated and characterized systematically. Results showed that Er had been successfully doped into a PbO_2_ catalyst layer and its doping smoothed the surface, increased electrode OEP, and enhanced the oxidation ability of PbO_2_, especially for 2% Er-PbO_2_. Electrochemical oxidation of SMR in Er-PbO_2_ system has been investigated. The pseudo first-order reaction kinetics constant obeyed the sequence of 2% Er-PbO_2_ > 4% Er-PbO_2_ > 1% Er-PbO_2_ > 0.5% Er-PbO_2_ > 0% PbO_2_. Effects of pH value and current density on SMR degradation have been analysed. The degradation pathway was proposed based on the active sites identified by DFT calculation and LC-MS analysis of the intermediates. All laboratorial results show that the electrochemical degradation of SMR using 2% Er-PbO_2_ electrodes as anodes has great potential for application. The efficient removal of SMR by this electrochemical system prevents both the enrichment of SMR in the human body through the biological chain and the creation of superbugs. This can effectively prevent the human body from developing resistance to sulfonamide antibiotics, which has great significance for the treatment of human bacterial infections and contributes to the maintenance of human health.

## Figures and Tables

**Figure 1 ijerph-19-13503-f001:**
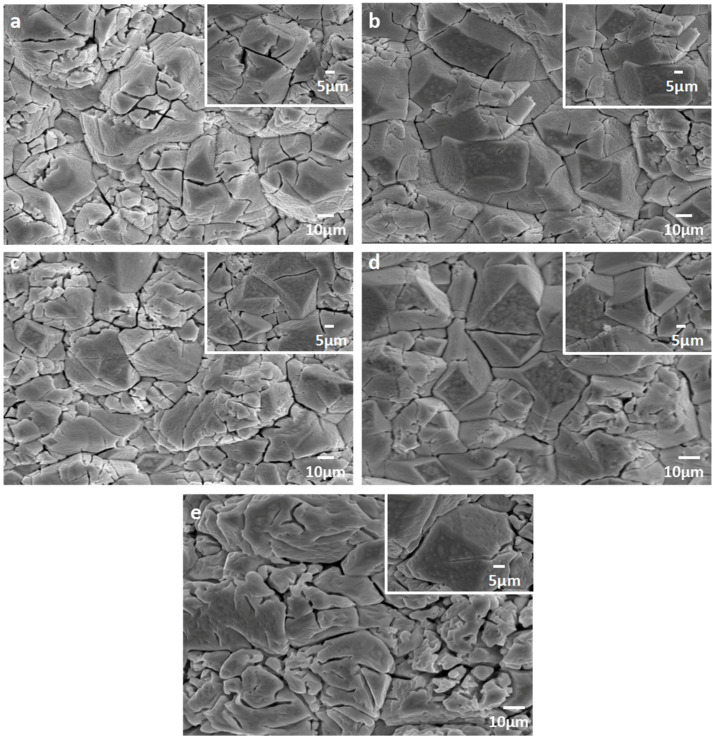
SEM diagram of distinct Er doped PbO_2_ electrodes: (**a**) (0% PbO_2_ electrode), (**b**) (0.5% Er-PbO_2_ electrode), (**c**) (1.0% Er-PbO_2_ electrode), (**d**) (2.0% Er-PbO_2_ electrode), (**e**) (4.0% Er-PbO_2_ electrode).

**Figure 2 ijerph-19-13503-f002:**
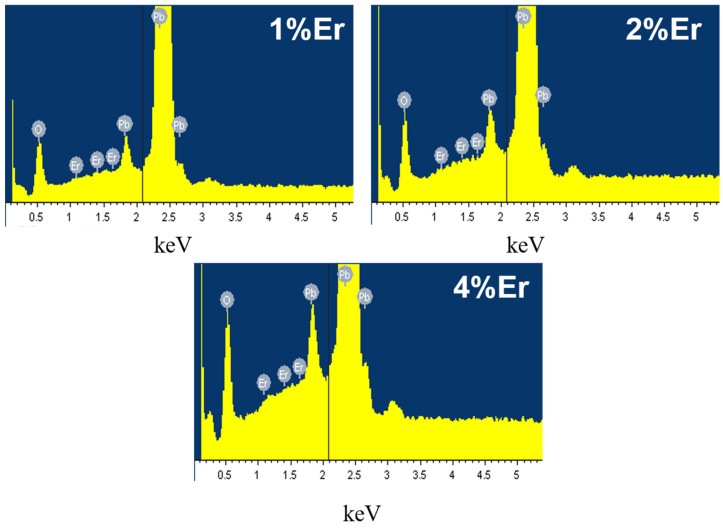
EDS spectra of different electrodes.

**Figure 3 ijerph-19-13503-f003:**
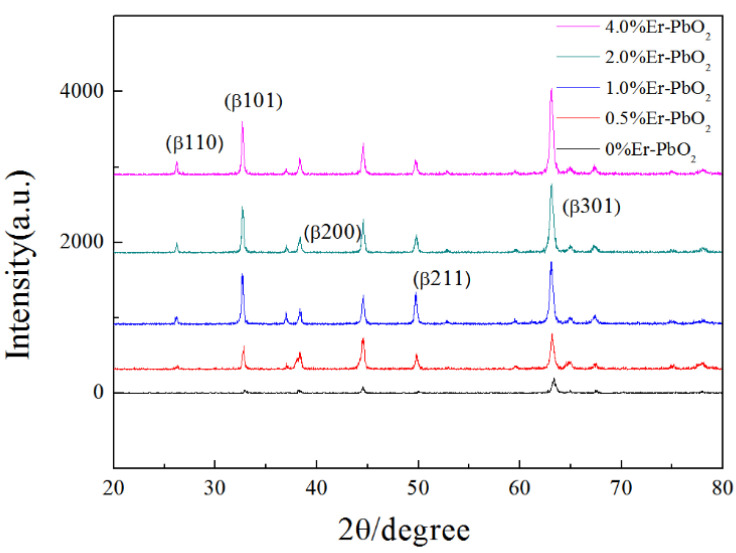
XRD pattern of distinct Er-PbO_2_ electrodes.

**Figure 4 ijerph-19-13503-f004:**
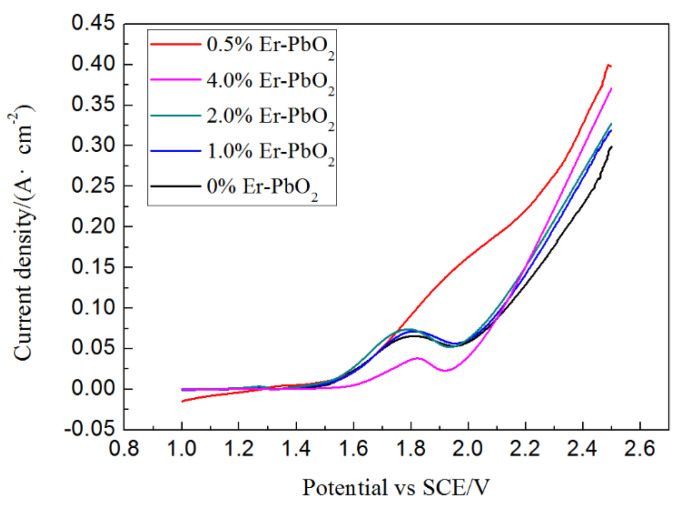
LSV of distinct Er-PbO_2_ measured in 0.5 M H_2_SO_4_ at 25 °C, scan rate: 10 mV·s^−1^.

**Figure 5 ijerph-19-13503-f005:**
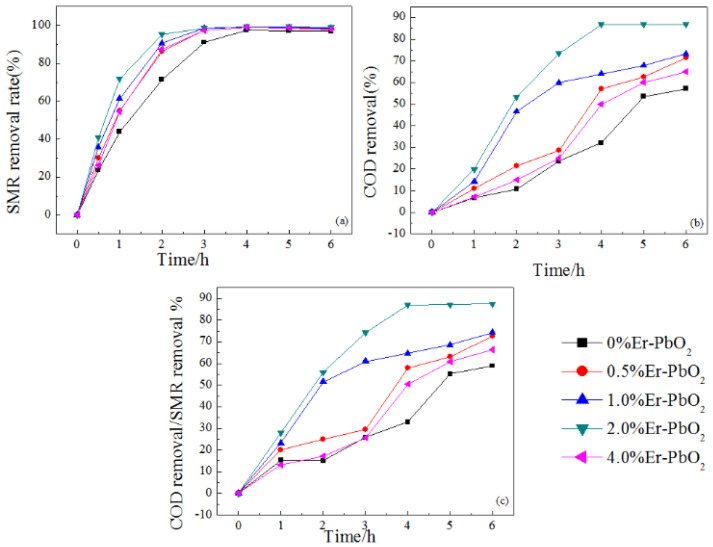
Removal rate of SMR (**a**) and COD removal rate (**b**), COD removal rate to SMR removal rate (**c**) in electrochemical degradation process of different electrodes.

**Figure 6 ijerph-19-13503-f006:**
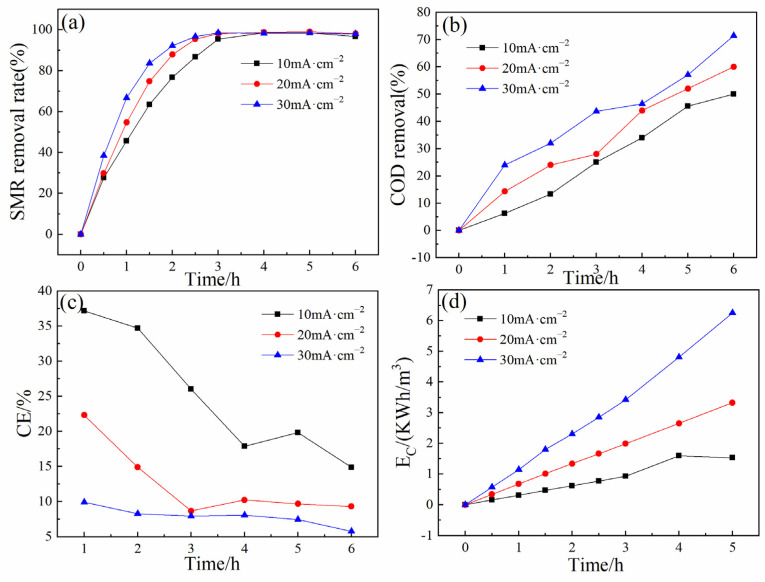
Removal rate of SMR at distinct current densities (**a**), removal rate of COD (**b**), current efficiency of electrochemical degradation of SMR (**c**), energy consumption (**d**).

**Figure 7 ijerph-19-13503-f007:**
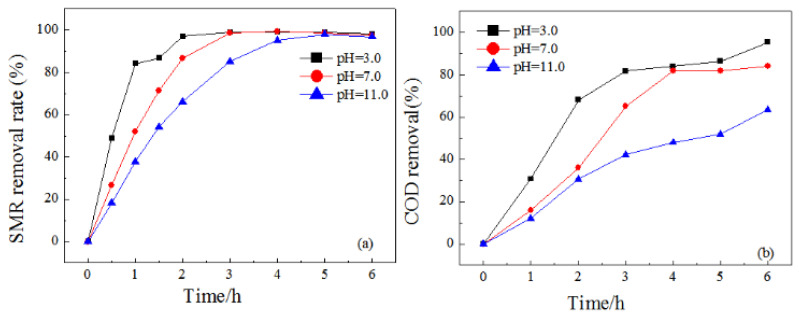
Removal rate of SMR (**a**), removal rate of COD (**b**) at different pH conditions.

**Figure 8 ijerph-19-13503-f008:**
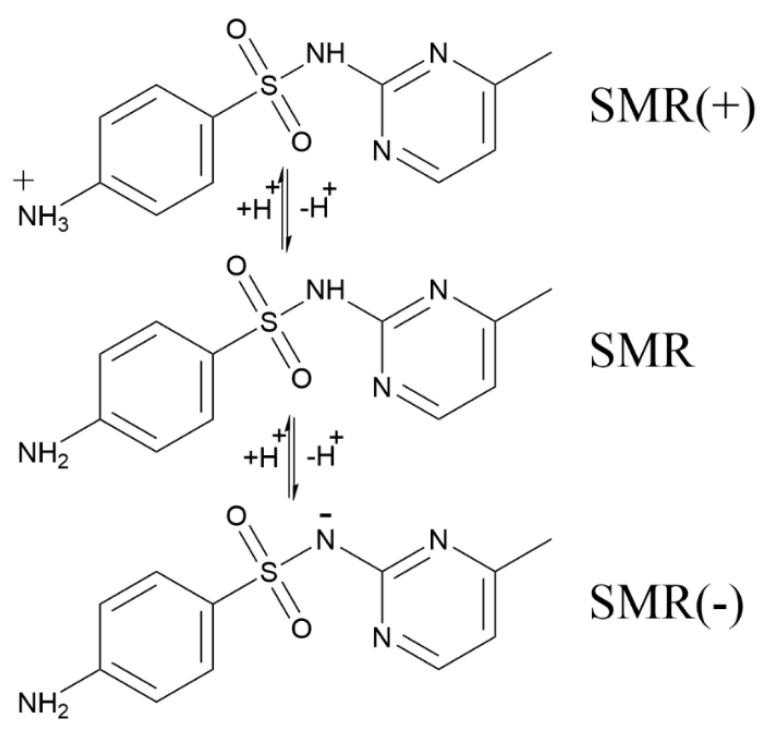
Hydrolysis structure of SMR.

**Figure 9 ijerph-19-13503-f009:**
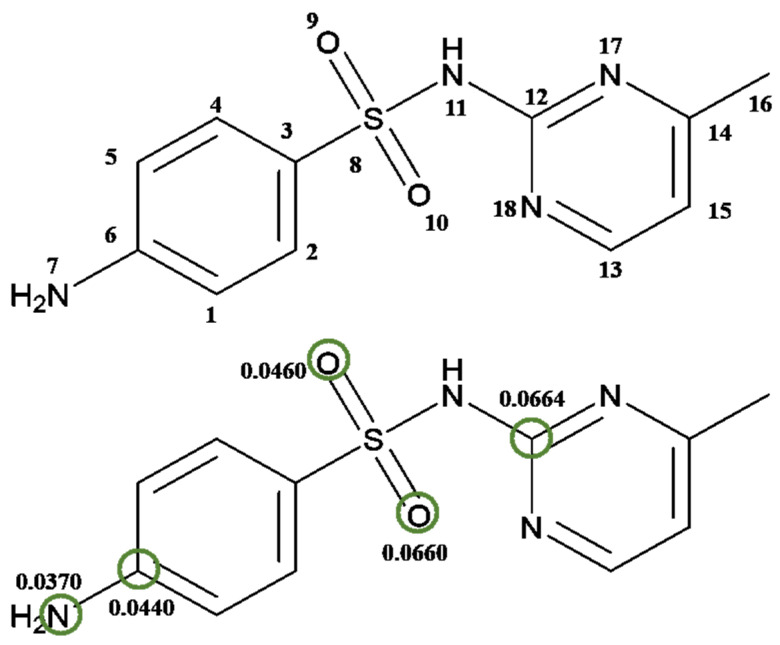
(**top**) Structure of SMR molecules. (**bottom**) possible active sites inferred from the Fukui index (*f* ^0^).

**Figure 10 ijerph-19-13503-f010:**
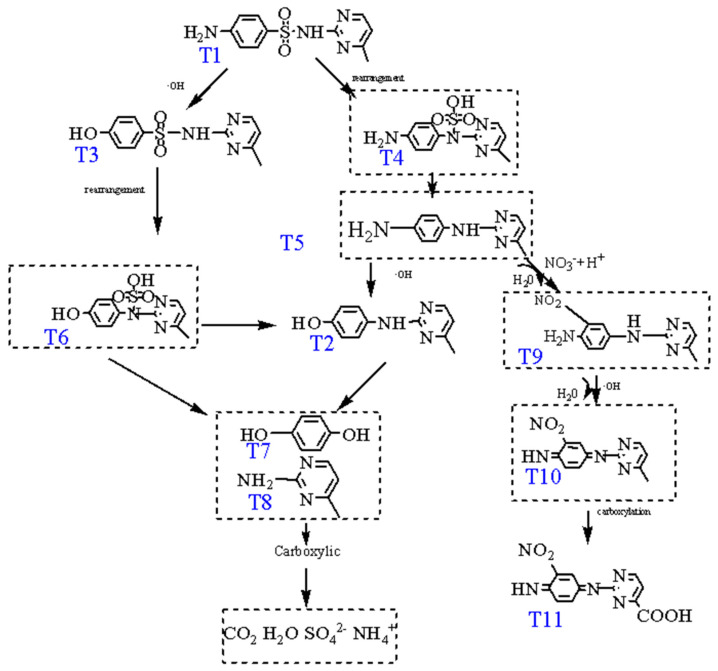
SMR degradation pathway of electrochemical oxidation.

**Table 1 ijerph-19-13503-t001:** Lattice dimensions of different Er-PbO_2_ electrodes.

Electrodes	WHM (301)	D (nm)
0% PbO_2_	0.39072	43.43
0.5% Er-PbO_2_	0.40659	41.73
1.0% Er-PbO_2_	0.42429	39.99
2.0% Er-PbO_2_	0.42462	39.96
4.0% Er-PbO_2_	0.40481	41.92

**Table 2 ijerph-19-13503-t002:** Fukui index ($f_{k}^{0}$) distribution on SMX.

Atom	q_k_ (N − 1)	q_k_ (N)	q_k_ (N + 1)	$f_{k}^{0}$
1C	−0.084593	−0.118527	−0.111559	0.013483
2C	−0.044314	−0.056261	−0.095393	0.0255395
3C	−0.180775	−0.199223	−0.182452	0.0008385
4C	−0.069064	−0.080553	−0.098221	0.0145785
5C	−0.079208	−0.114647	−0.115776	0.018284
6C	0.343255	0.298817	0.255246	**0.0440045**
7N	−0.582625	−0.655931	−0.656688	**0.0370315**
8S	1.285632	1.242465	0.925813	0.1799095
9O	−0.500962	−0.551447	−0.592888	**0.045963**
10O	−0.467286	−0.517021	−0.599261	**0.0659875**
11N	−0.681021	−0.698172	−0.597924	−0.0415485
12C	0.666093	0.652561	0.533312	**0.0663905**
13C	0.133549	0.124984	0.113774	0.0098875
14C	0.300235	0.297406	0.301063	−0.000414
15C	−0.118096	−0.139383	−0.157266	0.019585
16C	−0.370212	−0.363920	−0.363191	−0.0035105
17N	−0.490604	−0.516130	−0.528766	0.019081
18N	−0.476218	−0.468657	−0.448271	−0.0139735

Bold data: sites vulnerable to free radical attack.

**Table 3 ijerph-19-13503-t003:** m/z chart for the analysis of intermediates by LC-MS.

Number	Molecular Formula	m/z	Molecular Weight	Molecular Structure
T1	C_11_H_12_N_4_O_2_S	265.0756	264	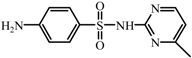
T2	C_11_H_11_N3O	202.0982	201	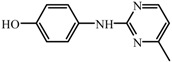
T3	C_11_H_11_N_3_O_3_S	266.0775	265	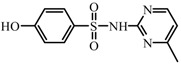
T11	C_11_H_7_N_5_O_4_	274.2759	273	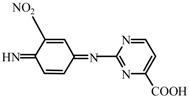

## Data Availability

Not applicable.

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
