# Peer review of "Fabrication of PbO2 Electrodes with Different Doses of Er Doping for Sulfonamides Degradation"

_ijerph, 2022, doi:10.3390/ijerph192013503_

Round 1
Reviewer 1 Report
Dear authors,
The manuscript is well organised and reveals an interesting and important solution to antibiotics degradation, avoiding their accumulation on the environment. It was a pleasure to read the manuscript. However, some important issues are missing. I suggest the authors to include in the Introduction section, information about antibiotics and their importance on the One Health perspective.
This manuscript is well written however English editing and complement the Conclusion section will be crucial to improve the quality of the manuscript.
Apart from the suggestions some mistakes were detected:
- Abstract section: I suggest to use SEM-EDS in the entire manuscript. Add space after “%” (2%Er-PbO2 > 4%Er-PbO2 > 1%Er-PbO2 > 0.5%Er-PbO2 > 0%PbO2).
Page 1, Line 14: Suggestion “….inducing resistance mechanisms on microorganisms”.
Page 2, Line 31: correct element to elements.
Page 2, Line 43: Suggestion “scanning electronic microscopy with energy dispersive spectroscopy (SEM-EDS).
Page 2, Line 45-47: Please clarify the sentence “In addition, possible degradation pathway of SMR was proposed planned supported active site identification by density function theory (DFT) calculation and intermediates detection by LC-MS.”.
Page 2, Line 48: Correct Experimental, eliminate or add the chemical structure for all the chemicals and add/remove some spaces in the experimental section, like Line 62, 64, 65.
Page 3, Line 78-79: Rewrite as “The synthetic acidic solution…”
Page 3, Line 84: SEM-EDS
Page 3, Line 92: Please specify time intervals (section 2.4).
Page 3; Section 2.5: In my opinion, it doesn't make sense a topic with few information. if the authors decide to maintain, change the title, once SEM-EDS and XRD are also analytical methods, and add some experimental details.
Page 3, Line 101: The authors mention “According to previous research…”, please add references.
Page 3, Line 105: I suggestion a small alteration of “These consequences” to “These results”.
Page 3, Section 3.1.2: Title suggestion “SEM-EDS analyzes” and sentence alteration “SEM-EDS (Fig. 1 and 2)”. Use analyzes in the following sections instead of analyzation.
Page 4, Line 118: Correct to “Er is present on electrode…”
Page 4, Figure 1: Please indicate the area on the 10um image of the zoomed image (5 um).
Page 5, Figure 2: Please edit Fig. 2 as Fig.1: a (0%PbO2 electrode), b (0.5%Er-120 PbO2 electrode), c (1.0%Er-PbO2 electrode), d (2.0%Er-PbO2 electrode), e (4.0%Er-PbO2 electrode).
Page 5, Line 128: Correct to “These results”
Page 6, Line 131: Pb4+
Page 6, Figure 3: Please identify each spectrum with a, b, ..... and complete the caption as previous figures (a (0%PbO2 electrode), b (0.5%Er-120 PbO2 electrode), c (1.0%Er-PbO2 electrode), d (2.0%Er-PbO2 electrode), e (4.0%Er-PbO2 electrode).
Page 6, Figure 4: I understand the idea, but I think that is better to sort graph lines by %, please.
Page 7, Figure 5: Please increase (a), (b) and (c) in this figure and also in figure 6 and 7.
Page 12, Line 168: I suggest to rewrite an impactful final remark, evidencing the benefits of this electrode to reduce/degrade SMR content, highlighting their contribution from One Health perspective.
Author Response
RESPONSE TO REVIEWER#1
General Comments:
The manuscript is well organized and reveals an interesting and important solution to antibiotics degradation, avoiding their accumulation on the environment. It was a pleasure to read the manuscript. However, some important issues are missing. I suggest the authors to include in the Introduction section, information about antibiotics and their importance on the One Health perspective.
Response: Many thanks for your valuable suggestions. The Introduction of this paper has been improved and highlighted in the revised manuscript.
Specific Comments:
(1) Abstract section: I suggest to use SEM-EDS in the entire manuscript. Add space after “%” (2% Er-PbO2 > 4% Er-PbO2 > 1% Er-PbO2 > 0.5% Er-PbO2 > 0% PbO2).
Response: Thank you for your valuable suggestions. SEM-EDS has been used in the entire manuscript and space has been added after “%”.
Abstract: “Surface morphology characterization by SEM-EDS and XRD showed that Er has been successfully doped into PbO2 catalyst layer and the particle size of Er-PbO2 was reduced significantly. Electrochemical oxidation of Sulfamerazine (SMR) in Er-PbO2 anode system obeyed pseudo-first order kinetic model with the order of 2% Er-PbO2 > 4% Er-PbO2 > 1% Er-PbO2 > 0.5% Er-PbO2 > 0% PbO2.”
(2) Page 1, Line 14: Suggestion “….inducing resistance mechanisms on microorganisms”.
Response: Thank you for your valuable suggestions. This sentence has been revised.
P1, L9-12: “The massive proliferation of antibiotics threatened environmental safety and human health even at an occasional level, by inducing resistance mechanisms on microorganisms.”
(3) Page 2, Line 31: correct element to elements.
Response: Thank you for your valuable suggestions. This error has been revised.
(4) Page 2, Line 43: Suggestion “scanning electronic microscopy with energy dispersive spectroscopy (SEM-EDS).
Response: Thank you for your valuable suggestions. This sentence has been revised.
L48-50: “The Er-PbO2 electrode was characterized by X-ray diffraction (XRD) and scanning electronic microscopy with energy dispersive spectroscopy (SEM-EDS), systematically.”
(5) Page 2, Line 45-47: Please clarify the sentence “In addition, possible degradation pathway of SMR was proposed planned supported active site identification by density function theory (DFT) calculation and intermediates detection by LC-MS.”
Response: Thank you for your valuable suggestions. This sentence has been revised. The original meaning of this statement is to determine the active sites of SMR by DFT calculation and LC-MS intermediate detection.
L52-53: “In addition, the possible degradation pathways of SMR were analyzed by density function theory (DFT) calculation and LC-MS intermediates detection.”
(6) Page 2, Line 48: Correct Experimental, eliminate or add the chemical structure for all the chemicals and add/remove some spaces in the experimental section, like Line 62, 64, 65.
Response: Thank you for your valuable suggestions. These errors have been revised.
(7) Page 3, Line 78-79: Rewrite as “The synthetic acidic solution…”
Response: Thank you for your valuable suggestions. This sentence has been revised.
L90-91: “The synthetic acidic solution consist of 0.5 M Pb(NO3)2 ,0.05 M NaF and 1.8 mL PTFE in 1 M HNO3.”
(8) Page 3, Line 84: SEM-EDS
Response: Thank you for your valuable suggestions. This sentence has been modified.
L95-96: “In this experiment, the element composition and surface morphology of the electrode were inspected by SEM-EDS (Japanese Hitachi s4800).”
(9) Page 3, Line 92: Please specify time intervals (section 2.4).
Response: Thank you for your valuable suggestions. The degradation lasted for six hours, and samples were taken by one hour intervals, as shown in the degradation map coordinate system
(10) Page 3; Section 2.5: In my opinion, it doesn't make sense a topic with few information. if the authors decide to maintain, change the title, once SEM-EDS and XRD are also analytical methods, and add some experimental details.
Response: Thank you for your valuable suggestions. The title was changed to analysis and calculation methods. This part includes HPLC detection conditions and calculation methods of CE and EC. All the changes have been highlighted in the revised manuscript (L109-142).
(11) Page 3, Line 101: The authors mention “According to previous research…”, please add references.
Response: Thank you for your valuable suggestions. References have been added.
[33] Gao, B; Sun, M; Ding, W; Ding, Z; Liu, W. Decoration of γ-graphyne on TiO2 nanotube arrays: Improved photoelectrochemical and photoelectrocatalytic properties. Appl. Catal. B-Environ. 2021, 281: 119492. [sciencedirect]
(12) Page 3, Line 105: I suggestion a small alteration of “These consequences” to “These results”.
Response: Thank you for your valuable suggestions. This sentence has been revised.
L151-152: “These results indicate that the conversion of Ti-based oxygen to TiO2-NCs can effectively extend the lifetime of the electrode.”
(13) Page 3, Section 3.1.2: Title suggestion “SEM-EDS analyzes” and sentence alteration “SEM-EDS (Fig. 1 and 2)”. Use analyzes in the following sections instead of analyzation.
Response: Thank you for your valuable suggestions. These problems have been revised.
(14) Page 4, Line 118: Correct to “Er is present on electrode…”
Response: Thank you for your valuable suggestions. This error has been revised.
(15) Page 4, Figure 1: Please indicate the area on the 10 μm image of the zoomed image (5 μm).
Response: Thank you for your valuable suggestions. It’s a pity that during SEM detection, we didn’t mark the exact position of pictures with scale of 5 μm and can’t indicate them on pictures with scale of 10 μm. However, the difference between pictures with scales of both 5 μm and10 μm can be distinguished, which indicated that different dose of Er doping effected characters of PbO2 electrodes.
(16) Page 5, Figure 2: Please edit Fig. 2 as Fig.1: a (0%PbO2 electrode), b (0.5%Er-120 PbO2 electrode), c (1.0%Er-PbO2 electrode), d (2.0%Er-PbO2 electrode), e (4.0%Er-PbO2 electrode).
Response: Thank you for your valuable suggestions. Fig. 2 has been revised.
(17) Page 5, Line 128: Correct to “These results”
Response: Thank you for your valuable suggestions. This error has been revised.
(18) Page 6, Line 131: Pb4+
Response: Thank you for your valuable suggestions. This error has been revised.
(19) Page 6, Figure 3: Please identify each spectrum with a, b, ..... and complete the caption as previous figures (a (0%PbO2 electrode), b (0.5%Er-PbO2 electrode), c (1.0%Er-PbO2 electrode), d (2.0%Er-PbO2 electrode), e (4.0%Er-PbO2 electrode).
Response: Thank you for your valuable suggestions. Fig. 3 has been revised.
(20) Page 6, Figure 4: I understand the idea, but I think that is better to sort graph lines by %, please.
Response: Thank you for your valuable suggestions. Fig. 4 has been revised.
(21) Page 7, Figure 5: Please increase (a), (b) and (c) in this figure and also in figure 6 and 7.
Response: Thank you for your valuable suggestions. Fig. 5, Fig. 6 and Fig. 7 has been revised.
(22) Page 12, Line 168: I suggest to rewrite an impactful final remark, evidencing the benefits of this electrode to reduce/degrade SMR content, highlighting their contribution from One Health perspective.
Response: Thank you for your valuable suggestions. This section has been revised.
L312-319: “The degradation pathway was proposed based on the active sites identified by DFT calculation and LC-MS analysis of the intermediates. All laboratorial results show that the electrochemical degradation of SMR using 2% Er-PbO2 electrodes as anodes has great potential for application. The efficient removal of SMR by this electrochemical system prevents both the enrichment of SMR in the human body through the biological chain and the creation of superbugs. This can effectively prevent the human body from developing resistance to sulfonamide antibiotics, which has great significance for the treatment of human bacterial infections and contributes to the maintenance of human health.”

Reviewer 2 Report
The article "Fabrication of PbO2 Electrodes with Different Doses of Er Doping for Sulfonamides Degradation" presents PbO2 electrodes doped with different doses of the rare earth element Er for the degradation of Sulphonamide. The samples were characterised using SEM and XRD, and the effects of various operational parameters were also investigated. Further, DFT was used to propose the degradation pathway.
However, I have a few questions/suggestions/comments regarding this study:
1) There are plenty of articles based on the doped variants of PbO2 in the literature. The authors haven't discussed any of them. The results obtained in this study has to be compared with those doped with Er and other rare-earth elements and explain the merit o their study.
2) The introduction should be rewritten based on these studies.
Few examples of Er doped PbO2 articles are: https://doi.org/10.1016/j.electacta.2019.135535 , https://doi.org/10.1016/j.electacta.2007.09.003, https://doi.org/10.1016/j.cclet.2020.03.073
There are plenty of other similar article available in the literature. As the authors stated, "Er doped in PbO2 electrode hasn’t been investigated sufficiently and the research limitation should be remedied" is not right.
3) The authors need to explain the novelty of their study based on the above articles.
4) There are lot of language/type errors in the manuscript.
5) The characterization methods presented here are not enough to confirm the doping percentage in the samples. Any other methods like XPS has to be performed.
6) How did the authors calculate the doping percentage?
7) Several cracks can be seen in the SEM images of all the samples. It is difficult to differentiate the reduction of cracks in some samples.
8) XRD is not a confirmative study for the particle size. It gives the average grain size. SEM presented here also does not confirm the same. Methods like TEM/SAED can confirm the particle size and the presence of Er.
9) As the authors say, there is no substantial difference in the particle/grain size as it can be seen in table 1.
10) Why is the particle size first decreasing for 0% to 2% Er doping and then increases for the sample with 4% doping?
11) Why is the presence of Er (even at 4% doping) not detected in the XRD, at least by a slight shift in the peak position? The ionic radii of Er is high.
12) The percentage of each elements have to presented with the EDS spectra.
13) Although the particle size decreases from 0% to 2% doping and then increases for 4% doping, why "The OEP increased in the order of PbO2, 1% Er-PbO2, 4%Er-PbO2, 0.5%Er-PbO2 and 2%Er-PbO2 electrodes"? Is it not dependant on the particle sizes? If yes, explain. If not, what else?
Author Response
RESPONSE TO REVIEWER#2
Specific Comments:
(1) There are plenty of articles based on the doped variants of PbO2 in the literature. The authors haven't discussed any of them. The results obtained in this study has to be compared with those doped with Er and other rare-earth elements and explain the merit of their study.
Response: Thank you for your valuable suggestions. The introduction has been revised and relevant discussions have been added. All of these changes have been highlighted in revised manuscript.
(2) The introduction should be rewritten based on these studies.
Few examples of Er doped PbO2 articles are: https://doi.org/10.1016/j.electacta.2019.135535, https://doi.org/10.1016/j.electacta.2007.09.003, https://doi.org/10.1016/j.cclet.2020.03.073
There are plenty of other similar article available in the literature. As the authors stated, "Er doped in PbO2 electrode hasn’t been investigated sufficiently and the research limitation should be remedied" is not right.
Response: Thank you for your valuable suggestions. This paper introduction has been rewritten and references has been added into introduction.
(3) The authors need to explain the novelty of their study based on the above articles.
Response: Thank you for your valuable suggestions. Related discussion and novelty of this article has been added in the revised manuscript.
Compared with previous research, Li Shuanghui [3] introduced Er3+ ions into SnO2 lattice to synthesize Er3+-SnO2 nanobelts via thermal evaporation methodology, which obviously improved the sensing performance of SnO2 electrodes. This research confirmed the great effect of Er in SnO2 electrode preparation and the urgency for research about Er doped PbO2 electrode preparation.
Wang Yanping [4] investigated electrochemical degradation of SMX by Ti/SnO2-Sb/Er-PbO2 electrode. Zhou Yuanzhen[5] investigated the electrochemical degradation of methylene blue by Ti/Sb2O3-SnO2/Er-PbO2 anodes. Both of these research focused on electrodes’ performance with little attention on the character of doping electrodes and the suitable doping dose investigation.
Based on these studies, we investigated the effects of different Er doping dose on character and oxidation ability of PbO2 electrode systematically, which remedied the research limitation. In addition, SMR was selected as the target pollutant, which has great threat on environmental healthy and is more difficult to be removed.
(4) There are lot of language/type errors in the manuscript.
Response: Thank you for your valuable suggestions. Many language/type errors have been corrected in the revised manuscript.
(5) The characterization methods presented here are not enough to confirm the doping percentage in the samples. Any other methods like XPS has to be performed.
Response: Thank you for your valuable suggestions. In the present work, different amount of Er(NO3)3 •5H2O were added during electrode preparation. Percentage of Er in PbO2 electrodes was determined by EDS. Other analyzation methods were not applied in the study because EDS is a hard evidence for Er doping.
(6) How did the authors calculate the doping percentage?
Response: Thank you for your valuable suggestions. The percentage of Er doping was determined by the amount of Er(NO3)3 •5H2O added in electrode preparation process. The percentage of Er doped into PbO2 electrode was calculated by EDS.
(7) Several cracks can be seen in the SEM images of all the samples. It is difficult to differentiate the reduction of cracks in some samples.
Response: Thank you very much for your valuable suggestions. It is true that the reduction of cracks in some samples is difficult to be distinguished. Therefore, Scherrer Equation was applied in the present study for average particle size calculation. According to Table 1, it was obvious that average particle size decreased along with Er doping (from 0% to 2%), while increased with Er doping increased to 4%. Thus, the suitable doping dose was identified to be 2%.
(8) XRD is not a confirmative study for the particle size. It gives the average grain size. SEM presented here also does not confirm the same. Methods like TEM/SAED can confirm the particle size and the presence of Er.
Response: Thank you for your valuable suggestions. Through SEM results, it can be found that particle size of Er doped PbO2 electrodes were various even on the same sample. Therefore, it is different to determine the exact particle size for electrodes by SEM, TEM or SAED. According to XRD analyzation, the average crystallite size (D) was calculated using Debye Scherrer’s formula [1], which could represent effect of Er doping in PbO2 electrodes. The presence of a small amount of Er can be seen in the EDS pattern.
(9) As the authors say, there is no substantial difference in the particle/grain size as it can be seen in table 1.
Response: Many thanks for your valuable questions. Doped Er replaced the position of Pb4+ in lattice but the doping dose wsn’t large enough to change the lattice structure (0% - 4%). Therefore, the particle size wasn’t substantial different [2]. However, the difference between electrode character, such as OEP, oxidation ability as well as particle size still can be distinguish, which demonstrated that Er doping effected the character of PbO2 electrode.
(10) Why is the particle size first decreasing for 0% to 2% Er doping and then increases for the sample with 4% doping?
Response: Many thanks for your valuable questions. Because of the special 4f orbital structure and excellent catalytic property of Er element, doping of Er could change the character of PbO2 electrode and improve its oxidation ability. Suitable dose of Er doping could affect the nucleation and crystallization during PbO2 preparation process. However, high dose of doping could inhibit the process. Therefore, the particle size decreased with doping dose form 0% to 2% and then increased with 4% doping.
(11) Why is the presence of Er (even at 4% doping) not detected in the XRD, at least by a slight shift in the peak position? The ionic radii of Er is high.
Response: Thank you for your valuable question. The missing of Er in XRD is mainly due to the low dose of doping. The XRD would be changed when dose of Er doping is large enough to change the lattice structure [2]. In the present study, the highest doping dose of Er is only 4%, which isn’t large enough to effect the lattice structure. Therefore, Er isn’t detected by XRD. According to the results of EDS detection, it can be found that Er is doped into PbO2 electrode and the character of electrode has been changed with Er doping.
(12) The percentage of each elements have to presented with the EDS spectra.
Response: Thank you for your valuable suggestions. Fig.2 shows the EDS spectra of different electrodes.
(13) Although the particle size decreases from 0% to 2% doping and then increases for 4% doping, why "The OEP increased in the order of PbO2, 1% Er-PbO2, 4%Er-PbO2, 0.5%Er-PbO2 and 2%Er-PbO2 electrodes"? Is it not dependant on the particle sizes? If yes, explain. If not, what else?
Response: Many thanks for your valuable suggestions. With different dose of Er doping, the characters of Er-PbO2 were various. For example, the particle size decreased with Er doping from 0% to 2% with the smallest particle size obtained at the doping dose of 2%. However, when the Er increased to 4%, particle size increased again. This phenomenon was consistent with changes of OEP, which achieved the highest value at doping dose of 2%. All of these results demonstrated that Er doping could enhance catalyst ability of PbO2 but the doping dose should be controlled at a certain range.
References
[1] Bulánek, R.; Hrdina, R.; Hassan, A.F. Preparation of polyvinylpyrrolidone modified nanomagnetite for degradation of nicotine by heterogeneous Fenton process. J. Environ. Chem. Eng. 2019, 7, 102988. [Sciencedirect]
[2] Zhou, Y.Z.; Li, Z.L.; Hao, C.T.; Zhang, Y.C.; Chai, S.N.; Han, G.P.; Xu, H.N.; Lu, J.S.; Dang, Y.; Sun, X.Q.; et al. Electrocatalysis enhancement of alpha, beta-PbO2 nanocrystals induced via rare earth Er(III) doping strategy: Principle, degradation application and electrocatalytic mechanism. Electrochim Acta. 2020, 333, 12. [sciencedirect]
[3] Li, S.H.; Liu, Y.K.; Wu, Y.M.; Chen, W.W.; Qin, Z.J.; Gong, N.L.; Yu, D.P. Highly sensitive formaldehyde resistive sensor based on a single Er-doped SnO2 nanobelt. Physica B. 2016, 489. [sciencedirect]
[4] Wang, Y.; Zhou, C.; Wu, J.; Niu, J. Insights into the electrochemical degradation of sulfamethoxazole and its metabolite by Ti/SnO2-Sb/Er-PbO2 anode. Chinese Chemical Letters. 2020, 31, 2673-2677. [sciencedirect]
[5] Zhou, Y.Z.; Li, Z.L.; Hao, C.T.; Zhang, Y.C.; Chai, S.N.; Han, G.P.; Xu, H.N.; Lu, J.S.; Dang, Y.; Sun, X.Q.; et al. Electrocatalysis enhancement of alpha, beta-PbO2 nanocrystals induced via rare earth Er(III) doping strategy: Principle, degradation application and electrocatalytic mechanism. Electrochim Acta. 2020, 333, 12. [sciencedirect]

Reviewer 3 Report
This manuscript examines the modification of PbO2 electrode via Er doping and the application for the destruction of antibiotics (SMR). The topic belongs to IJERPH , however, the manuscript needs major revision before publication.
Specific points
Figure 4 and paragraph 3.1.4: Why the OEP increased in the order of PbO2, 1%Er-PbO2, 4%Er-PbO2, 0.5%Er-PbO2 and 2%Er-PbO2 and not 0-0.5-1-2-4 for example?
3.2: The authors provide us the kinetics equation where t=time, but they display the time in hours and write the estimated kinetic constant without units inside the text, supplementary material and the abstract …
It is not clear to me how the authors estimate table S1. More details are needed.
The concentration SMR examined limits the environmental importance of the work. However, I understand that allows the estimation of COD . On the other side, the manuscript lacks the environmental part, while the journal in question is IJERPH. For example, the authors must demonstrate us some results in a real matrix (i.e, surface or groundwater spiked with SMR) or, more meaningful, use different electrolytes like NaCL to investigate also the efficiency of the system in the presence of chlorides that usually change the mechanism.
What is the price of Erb in relation to PbO2? From an engineering perspective, is it worth using Erb to enhance efficiency?
Author Response
RESPONSE TO REVIEWER#3
Specific Comments:
(1) Figure 4 and paragraph 3.1.4: Why the OEP increased in the order of PbO2, 1%Er-PbO2, 4%Er-PbO2, 0.5%Er-PbO2 and 2%Er-PbO2 and not 0-0.5-1-2-4 for example?
Response: Many thanks for your valuable suggestions. With different dose of Er doping, the characters of Er-PbO2 were various. For example, the particle size decreased with Er doping from 0% to 2% with the smallest particle size obtained at the doping dose of 2%. However, when the Er increased to 4%, particle size increased again. This phenomenon was consistent with changes of OEP, which achieved the highest value at doping dose of 2%. All of these results demonstrated that Er doping could enhance catalyst ability of PbO2 but the doping dose should be controlled at a certain range.
(2) The authors provide us the kinetics equation where t=time, but they display the time in hours and write the estimated kinetic constant without units inside the text, supplementary material and the abstract …
Response: Thank you for your valuable suggestions. This error has been revised.
L204-208: “The first-order kinetic fitting of the electrochemical decomposition process was shown in Eq. (4).
ln(C0/C) = kt (4)
where C0 and C wais concentration of SMR (mg L-1) at electrolysis time of 0 and t (h), k was the reaction rate constant (h-1).”
(3) It is not clear to me how the authors estimate table S1. More details are needed.
Response: Thank you for your valuable suggestions.
Accelerated lifetime was performed by anodic polarization of the different electrodes at 1 A cm-2 in a 0.2 M H2SO4 solution. The anode potential was measured as function of time and the electrode was considered to be deactivated when the potential increased to 10 V from its initial value. The lifetime practical application has been calculated according to Eq. (1):
t = (A1/A)2 t1 (1)
where A1 is the current density in the accelerated test (1 A/cm2), A is the current density in practical applications (20 mA cm-2), t1 is the lifetime of the electrode in the accelerated test (h), and t is the lifetime in practical applications (h).
Based on these results, lifetime of electrodes different substrates were calculated.
Table S1 Accelerate life time of prepared electrodes
|
|
Ti/Sb-SnO2/PbO2 |
TiO2-NCs/Sb-SnO2/PbO2 |
|
Accelerate life time/h |
10 |
35 |
|
Service lives/year |
2.85 |
9.99 |
(4) The concentration SMR examined limits the environmental importance of the work. However, I understand that allows the estimation of COD. On the other side, the manuscript lacks the environmental part, while the journal in question is IJERPH. For example, the authors must demonstrate us some results in a real matrix (i.e, surface or groundwater spiked with SMR) or, more meaningful, use different electrolytes like NaCl to investigate also the efficiency of the system in the presence of chlorides that usually change the mechanism.
Response: Thank you very much for your valuable suggestions. The concentration of sulfonamides on surface water is different in different position. According to Si Li’s research [1], the maximum concentration for SMR is 32.7 ng L-1 in Yangtze River and the average concentration of SMR is 1.62 ng L-1.
In addition, the effect of NaCl as electrolyte in electrochemical oxidation system has been investigate in our previous study. According to Hao Hai’s research [2], The pseudo-first-order kinetic constant for SMX degradation was 0.02 and 0.04 min-1 with Na2SO4 and NaCl under the same conditions (Fig. 1). For COD mineralization, after 6 h electrolysis, about 76.5% was removed when NaCl used as supporting electrolyte while only 65.7% was degraded in the system when Na2SO4 was used as supporting electrolyte. The enhancement of SMX degradation with NaCl as electrolyte was mainly due to the active chlorine (Cl2, HClO and ClO−) produced by direct oxidation of chlorine ions at anode surface (Eq. (2~4)) and indirect oxidation by ·OH to produce chlorine (Eq. (5~6)).
2Cl- → Cl2+ 2e− (2)
Cl2 + H2O → HClO + H+ + Cl- (3)
HClO → H+ + ClO- (4)
Cl- + ·OH → Cl· (5)
Cl·+ Cl· → Cl2 (6)
Fig.1 Effects of supporting electrolyte on SMX degradation (a) and COD evolution (b). Initial concentration of SMX was 30 mg L-1, Na2SO4, NaCl and NaNO3 was 0.1 M, respectively. Electrode gap was set to be 1 cm, T was 23 ± 1 ℃, volume of electrolyte was 350 mL, area of BDD electrode was 2 cm2 and pH was 7.
(5) What is the price of Er in relation to PbO2? From an engineering perspective, is it worth using Er to enhance efficiency?
Response: Thank you for your valuable suggestions.
In this paper, 2% Er-PbO2 electrode was prepared by adding 10mM Er(NO3)3 •5H2O($20.34/kg) which increased the anode cost by about 0.07$. However, the COD removal rate of 2% Er-PbO2 anode increased from 58.9% to 87.6% when treating SMR solution compared to 0% Er-PbO2 anode. Therefore, from an engineering point of view, it is feasible to use Er to improve electrode efficiency.
References:
[1] Li S., Shi W., Liu W., et al. A duodecennial national synthesis of antibiotics in China's major rivers and seas (2005-2016). Sci Total Environ., 615(2018) 906-917.
[2] Hai H., Xing X., et al. Electrochemical oxidation of sulfamethoxazole in BDD anode system: Degradation kinetics, mechanisms and toxicity evaluation. Sci. Total Environ., 738 (2020) 139909.

Reviewer 4 Report
The manuscript titled "Fabrication of PbO2 electrodes with different doses of Er Doping for Sulfonamides Degradation" has been reviewed and few comments are listed below and need major improvement before acceptance for the publication:
Comments:
1. Section 2.1: lack of information on the listed materials that has been used in this work.
2. Section 2.3: lack of information on the methods of characterization of the prepared electrodes.
3. Section 2.5: detail analytical technique should be in the main manuscript which is easier for the reader to relate all the important technique directly towards the discussion.
4. Basically, Sherrer equation in the XRD section was reported to determine the degree of crystallite size, however, author used this equation to report the particle size of the sample, which is not accurate – Author should use SEM or particle size analyser for determination of accurate particle size.
5. DFT calculation and few other analysis have been reported in the discussion part, however not mentioned in the methodology section. Please improve your methodology section thoroughly.
6. Lot of references are outdated and need to be updated.
7. Technical error can be easier detected in this manuscript, e.g., spacing, typo, formating, units and grammatical errors. Please correct all small mistakes.
Author Response
RESPONSE TO REVIEWER#4
Specific Comments:
(1) Section 2.1: lack of information on the listed materials that has been used in this work.
Response: Thank you for your valuable suggestions. 2.1 Chemicals information has been added.
L57-69: “All chemical reagents employed in the experiments were of analytical quality. Acetone, lead oxide, sodium hydroxide, lead nitrate, potassium dichromate, silver sulfate and mercury sulfate were purchased from traditional Chinese Medicine Group Chemical Reagent Co., Ltd. ammonium fluoride, hydrofluoric acid, nitric acid, phosphate, concentrated acid, n-butanol were purchased from Beijing Chemical Plant, antimony trichloride were bought from Tianjin Dachen Chemical Reagent, tin tetrachloride pentahydrate, sodium fluoride were bought from Tianjin Fuchen Chemical Reagent Plant. Ethylene glycol was obtained from Tianjin Zhiyuan Chemical Reagent Co., Ltd. nitrate bait was obtained from Shaanxi Ruikexin material Co., Ltd. polytetrafluoroethylene was obtained from Dongguan Jianyang Polymer material Co., Ltd. Sulfamerazine was obtained from Beijing Solaibao Technology Co., Ltd. Ultrapure water (Milli-Qρ=18.2 MΩ cm/25 ℃) was used throughout this study.”
(2) Section 2.3: lack of information on the methods of characterization of the prepared electrodes.
Response: Thank you for your valuable suggestions. 2.3 Electrode characterization has been added.
L96-98: “In this experiment, the element composition and surface morphology of the electrode were inspected by SEM-EDS (Japanese Hitachi s4800). The crystal phase composition and the grain size was analyzed by XRD, which was XD-DI type, using Cu Kα radiation (36 KV, 30 mA), Scanning speed is 4° min−1, 2θ = 20°-80°.”
(3) Section 2.5: detail analytical technique should be in the main manuscript which is easier for the reader to relate all the important technique directly towards the discussion.
Response: Thank you for your valuable suggestions. Detailed analysis techniques have been put into the main manuscript.
L120-136: “The linear sweep voltammetry experiment was executed to acquire their oxygen evolution overpotential at room temperature using a computerized electrochemical workstation (CHI 630E, Shanghai Chenhua, China) with a conventional three-electrode system, where the prepared electrodes served as working electrodes, while platinum sheet and a saturated calomel electrode (SCE) were used as the counter and reference electrodes, separately. The concentrations of SMR were measured by high performance liquid chromatography system (HPLC, LC-20A, Shimadzu Company, Japan). The mobile liquid phase was a mixed solution embracing 60% (by volume) methanol and 40% water. The separation was implemented using an Agilent SB-C18 (4.6 mm × 250 mm, 5 μm) at a pillar temperature of 25 ℃ and at a flow rate of 1 mL min-1, The UV detector wavelength was set at 265 nm. The injection volume was 25 μL. COD was determined according to the national standard method (GB11914-1989). The current efficiency was computed as Eq. (1):
(1)
where CODt and COD0 are the chemical oxygen demand (g L-1) at time t (s) and zero, separately, I is the current (A), t is the electrolysis time (s), F is the Faraday constant (96,287 C mol-1), and V is the electrolyte volume (L).
The energy consumption (EC) in the process of electrochemical oxidation was computed using Eq. (2):
(2)
where U is the average cell voltage (V), I is the current(A), t is the degradation time (h), V is the wastewater volume (m3).
The intermediates during the degradation of SMR were determined with LC-MS (LC-20ADXR,Shimadzu,Japan and API3200 Qtrap, Applied biosystems, USA).The mobile phase consisted of two solutions namely, A and B. Solution A was high pure water containing 0.01% formic acid , whereas solution B was methanol. The flow velocity was 0.4 mL min-1 and the temperature was kept at 40 ℃.”
(4) Basically, Sherrer equation in the XRD section was reported to determine the degree of crystallite size, however, author used this equation to report the particle size of the sample, which is not accurate – Author should use SEM or particle size analyser for determination of accurate particle size.
Response: Many thanks for your valuable suggestions. Accord to SEM results, it can be found that particle size is various even on the same PbO2 electrode. Therefore, it is difficult to determine a exact size for the electrode. Average particle size calculated by Sherrer equation also could reflect the electrode character.
(5) DFT calculation and few other analysis have been reported in the discussion part, however not mentioned in the methodology section. Please improve your methodology section thoroughly.
Response: Thank you for your valuable suggestions. Detailed DFT calculation method has been added.
L137-142: “The active point of SMR in the degradation process of electrocatalytic system was inferred by DFT calculation, and the related work was done by Gaussian 09 program. Optimization of SMR geometry at the atomic level of the DFT theory B3LYP/6-31G(d, p). Afterwards, the active sites of SMR molecules vulnerable to free radical attack were identified by calculating the Fukui function (Eq. (3)).
=[(qkN-1)- (qkN+1)]/2 (3)”
(6) Lot of references are outdated and need to be updated.
Response: Thank you for your valuable suggestions. Some references have been updated.
(7) Technical error can be easier detected in this manuscript, e.g., spacing, type, formating, units and grammatical errors. Please correct all small mistakes.
Response: Thank you for your valuable suggestions. Many language/type errors have been corrected based on your and other reviewers' suggestions.

Round 2
Reviewer 2 Report
The revised manuscript "Fabrication of PbO2 Electrodes with Different Doses of Er Doping for Sulfonamides Degradation" still have lot of issues. A few of the previous comments were not properly addressed. Please address the following comments:
1. English language needs thorough revision (eg. line 39-40, 157-158, etc). Please check the complete manuscript again.
2. How does the EDS spectra presented in the manuscript represent the precise doping percentage? An increase in Er percentage can be seen in the images. But any quantification can hardly be achieved from the EDS. Hence, it is better to try XPS, EPMA or SIMS.
3. The authors claim (in the abstract) that "Surface morphology characterization by SEM-EDS and XRD showed that Er has been successfully doped into PbO2 catalyst layer and the particle size of Er-PbO2 was reduced significantly"
It is to be noted that there is no substantial reduction in the particle size. As the authors claims in their review comments, it is true that the particle size varies in all the samples. Also, XRD analysis gives the average crystallite sizes, as evident from the SEM images too. SEM images display crystallites having sizes in the micrometer range. As the authors claim particle sizes less than 50 nm, it is better to get the TEM, plot the particle size histogram along with the SEM and XRD results.
4. Is there any reference for your comment "Suitable dose of Er doping could affect the nucleation and crystallization during PbO2 preparation process. However, high dose of doping could inhibit the process. Therefore, the particle size decreased with doping dose form 0% to 2% and then increased with 4% doping." ?
5. Answer to Q13 is not satisfactory. Plz see that the particle size decreases varies in the order 0%>0.5%>1%>2% <4%. But the OED increases in a different order (1% Er-PbO2< 4%Er-PbO2< 0.5%Er-PbO2 < 2%Er-PbO2). Why?
A major revision is recommended.
Author Response
RESPONSE TO REVIEWER#2
Specific Comments:
(1) English language needs thorough revision (eg. line 39-40, 157-158, etc). Please check the complete manuscript again.
Response: Thank you for your valuable suggestions. The English language problems in the relevant paragraphs have been modified.
L39-40: “Wang Yanping[1] prepared Ti/SnO2-Sb/Er-PbO2 electrode for electrochemical degradation of SMX.”
L157-158: “In addition, EDS analysis of different electrodes showed that Er element existed on the surface of electrodes.”
(2) How does the EDS spectra presented in the manuscript represent the precise doping percentage? An increase in Er percentage can be seen in the images. But any quantification can hardly be achieved from the EDS. Hence, it is better to try XPS, EPMA or SIMS.
Response: Thank you for your valuable suggestions. During electrode preparation, different doses of Er(NO3)3 •5H2O were added into electrolyte for electrodes of 0% Er-PbO2, 0.5% Er-PbO2, 1% Er-PbO2, 2% Er-PbO2, 4% PbO2, respectively. According to EDS results, weight and atomic percentage of Er in electrodes have been investigated and results have been shown in Table S2. When dose of Er increased from 0 to 4%, the atomic percentage of Er increased from 0 to 0.18. Although the doping percentage isn’t high, the rising tendency is significant. It is true that EPMA or SIMS is a useful method for Er quantification and we will applied them in the following studies. Many thanks for your valuable suggestions again.
Table S2 Percentage of different elements in Er-PbO2 by EDS
|
|
O |
Pb |
Er |
|||
|
|
Weight Percentage |
Atomic Percentage |
Weight Percentage |
Atomic Percentage |
Weight Percentage |
Atomic Percentage |
|
0% Er-PbO2 |
12.23 |
64.34 |
87.77 |
35.66 |
0 |
0 |
|
0.5% Er-PbO2 |
12.31 |
64.51 |
87.52 |
35.41 |
0.09 |
0.05 |
|
1% Er-PbO2 |
12.28 |
64.45 |
87.57 |
35.48 |
0.14 |
0.07 |
|
2% Er-PbO2 |
10.94 |
61.39 |
88.87 |
38.50 |
0.19 |
0.10 |
|
4% Er-PbO2 |
11.81 |
63.40 |
87.85 |
36.42 |
0.34 |
0.18 |
Figure S1 EDS results of different Er-PbO2 electrodes
(3) The authors claim (in the abstract) that "Surface morphology characterization by SEM-EDS and XRD showed that Er has been successfully doped into PbO2 catalyst layer and the particle size of Er-PbO2 was reduced significantly"
It is to be noted that there is no substantial reduction in the particle size. As the authors claims in their review comments, it is true that the particle size varies in all the samples. Also, XRD analysis gives the average crystallite sizes, as evident from the SEM images too. SEM images display crystallites having sizes in the micrometer range. As the authors claim particle sizes less than 50 nm, it is better to get the TEM, plot the particle size histogram along with the SEM and XRD results
Response: Thank you for your valuable suggestions.
During the research, we referred other researchers’ results. Zhou Yuanzhen[2] et al. doped Er(Ⅲ) into PbO2 lattice and characterized by SEM, EDS and XRD. The average grain size of PbO2 and Er-PbO2 calculated according to Debye Scherrer’s formula is 63 nm and 57 nm, respectively. Yijing Xia[3] et al. doped In into PbO2 lattice and The average crystal sizes calculated by Scherrer equation were 61.43 nm and 49.95 nm, respectively. Yingwu Yao[4] et al. investigated Yb doped PbO2 electrode. The crystal size for Yb-doped PbO2 electrodes is 14.9 nm calculated by Scherrer equation, which is finer than that for pure PbO2 electrodes as 20.1 nm. It should be noticed that Yijing Xia applied TEM to analyze the particle size of PbO2. However, the particles should be peeled off from the substrate and the agglomeration phenomenon effected the size identification.
Fig.SM-1. TEM of In-doped PbO2 film (a, b, c) and PbO2 film (d).[4]
Xiaoyue Duan[5] also analyzed CNT-Ce-PbO2 electrode by TEM. This method could help to characterize the morphology of electrode by can’t be used to identified the particle size.
Fig.2 TEM of CNT-Ce-PbO2 electrode[5]
Based on these research, we proposed to identified the particle size according to XRD results. In the following research, TEM will be applied to characterize morphology of PbO2 electrodes. Thank you for your kindly suggestions again.
(4) Is there any reference for your comment "Suitable dose of Er doping could affect the nucleation and crystallization during PbO2 preparation process. However, high dose of doping could inhibit the process. Therefore, the particle size decreased with doping dose from 0% to 2% and then increased with 4% doping."?
Response: Many thanks for your kindly suggestions. Some researches about different doping dose analyzation have been referred to explain the dose affection. Jianmin Chen[6] investigated different dose of Al3+ doping in PbO2 electrode. “The CAP removal capacity increased as the Al/Pb molar ratio in the electrodeposition solution increased from 0 to 1%. However, further increase of Al/Pb molar ratio reduced the CAP removal capacity. This result indicated that the introduction of Al3+ could improve the electrochemical activity of PbO2 electrode for the oxidation of organic compounds, however, too much Al3+ amount in the electrodeposition appeard to have an adverse effect on the electrochemical activity of PbO2 electrode.” Yujie Feng[7] investigated different dose of Gd (0%, 1%,2%, 3.3%, 5% and 10%) doping in PbO2 electrode and results showed that 2%Gd-PbO2 had the best catalyst ability. The oxidation ability of Gd-PbO2 increased with Gd dose increased form 0% to 2%, while decreased with the dose of Gd further increasing. Based on above results, catalyst ability increased with doping dose increasing at first, and then decreased with dose further increasing. All of them indicated that there was a suitable dose existed during the doping. In our research, the change trend of oxidation ability and particle size were consistent with this regulation.
(5) Answer to Q13 is not satisfactory. Plz see that the particle size decreases varies in the order 0%>0.5%>1%>2% <4%. But the OED increases in a different order (1% Er-PbO2< 4%Er-PbO2< 0.5%Er-PbO2 < 2%Er-PbO2). Why?
Response: Thank you for your valuable suggestions. The order of OEP and particle size is different. This result may be because OEP is not only related to particle size, but also due to the replacement of Pb4+ by Er, which resulted in crystal point defects [2]. In the research of Yb-doped PbO2 electrodes[4], the oxidation ability order 0 mmol < 1 mmol < 3 mmol >5 mmol < 7 mmol, which is also irregular. Based on these results, there is no doubt about the general trendy, but some irregular data would exist.
Fig. 1. (a) The effect of ytterbium ion concentrations on acetamiprid removal ratio, (b) the kinetic curves. (initial pH value 5.0; initial acetamiprid concentration 20 mg L-1; current density 150 mA cm-2). (■) 0 mmol L-1, (●) 1 mmol L-1, (▲) 3 mmol L-1, (▾) 5 mmol L-1, (◆) 7 mmol L-1.[4]
Reference
- Wang, Y.; Zhou, C.; Wu, J.; Niu, J. Insights into the electrochemical degradation of sulfamethoxazole and its metabolite by Ti/SnO2-Sb/Er-PbO2 anode. Chinese Chemical Letters. 2020, 31, 2673-2677. [sciencedirect]
- Zhou, Y.Z.; Li, Z.L.; Hao, C.T.; Zhang, Y.C.; Chai, S.N.; Han, G.P.; Xu, H.N.; Lu, J.S.; Dang, Y.; Sun, X.Q.; et al. Electrocatalysis enhancement of alpha, beta-PbO2 nanocrystals induced via rare earth Er(III) doping strategy: Principle, degradation application and electrocatalytic mechanism. Electrochim Acta. 2020, 333, 12. [sciencedirect]
- Xia, Y.J.; Bian, X.Z.; Xia, Y.; Zhou, W.; Wang, L.; Fan, S.Q.; Xiong, P.; Zhan, T.T.; Dai, Q.Z.; Chen, J.M. Effect of indium doping on the PbO2 electrode for the enhanced electrochemical oxidation of aspirin: An electrode comparative study. Sep. Purif. Technol. 2020, 237, 7. [sciencedirect]
- Yao Y.W, Teng G.G, Yang Y, et al. Electrochemical oxidation of acetamiprid using Yb-doped PbO2 electrodes: Electrode characterization, influencing factors and degradation pathways. Sep. Purif. Technol. 2019, 211: 456-466. [sciencedirect]
- Duan X.Y., Zhao Y.Y., Liu W., Chang L.M., Li X., Electrochemical degradation of p-nitrophenol on carbon nanotube and Ce modified-PbO2 electrode. J. Taiwan Ins. Chem. E. 2014, 45: 2975–2985.
- Chen J.M., Xia Y.J., Dai Q.Z., Electrochemical degradation of chloramphenicol with a novel Al doped PbO2 electrode: Performance, kinetics and degradation mechanism. Electrochim. Acta 2015, 165: 277–287.
- Feng Y.J., Cui Y.H., Bruce L., Liu Z.Q., Performance of Gd-doped Ti-based Sb-SnO2 anodes for electrochemical destruction of phenol. Chemosphere 2008, 70: 1629–1636.

Reviewer 3 Report
The authors revised their work according to the reviewers comments
Author Response
Many thanks for the reviewer's valuable suggestions. This manuscript has been revised according to the reviewers' comments.